

# Identification of prognostic risk factors for pancreatic cancer using bioinformatics analysis

Dandan Jin[1,2,*], Yujie Jiao[1,2,*], Jie Ji[1,2], Wei Jiang[3], Wenkai Ni[1], Yingcheng Wu[2], Runzhou Ni[1], Cuihua Lu[1], Lishuai Qu[1], Hongbing Ni[1], Jinxia Liu[1], Weisong Xu[4] and MingBing Xiao[1,5]

[1] Department of Gastroenterology, Affiliated Hospital of Nantong University, Nantong, China
[2] Clinical Medicine, Medical College, Nantong University, Nantong, China
[3] Department of Emergency, Affiliated Hospital of Nantong University, Nantong, China
[4] Department of Gastroenterology, Second People's Hospital of Nantong, Nantong, China
[5] Research Center of Clinical Medicine, Affiliated Hospital of Nantong University, Nantong, China
[*] These authors contributed equally to this work.

Corresponding authors
MingBing Xiao, xmb73@163.com
Weisong Xu, xws71@sina.com

## ABSTRACT

**Background**. Pancreatic cancer is one of the most common malignant cancers worldwide. Currently, the pathogenesis of pancreatic cancer remains unclear; thus, it is necessary to explore its precise molecular mechanisms.

**Methods**. To identify candidate genes involved in the tumorigenesis and proliferation of pancreatic cancer, the microarray datasets GSE32676, GSE15471 and GSE71989 were downloaded from the Gene Expression Omnibus (GEO) database. Differentially expressed genes (DEGs) between Pancreatic ductal adenocarcinoma (PDAC) and nonmalignant samples were screened by GEO2R. The Database for Annotation Visualization and Integrated Discovery (DAVID) online tool was used to obtain a synthetic set of functional annotation information for the DEGs. A PPI network of the DEGs was established using the Search Tool for the Retrieval of Interacting Genes (STRING) database, and a combination of more than 0.4 was considered statistically significant for the PPI. Subsequently, we visualized the PPI network using Cytoscape. Functional module analysis was then performed using Molecular Complex Detection (MCODE). Genes with a degree $\geq 10$ were chosen as hub genes, and pathways of the hub genes were visualized using ClueGO and CluePedia. Additionally, GenCLiP 2.0 was used to explore interactions of hub genes. The Literature Mining Gene Networks module was applied to explore the cocitation of hub genes. The Cytoscape plugin iRegulon was employed to analyze transcription factors regulating the hub genes. Furthermore, the expression levels of the 13 hub genes in pancreatic cancer tissues and normal samples were validated using the Gene Expression Profiling Interactive Analysis (GEPIA) platform. Moreover, overall survival and disease-free survival analyses according to the expression of hub genes were performed using Kaplan-Meier curve analysis in the cBioPortal online platform. The relationship between expression level and tumor grade was analyzed using the online database Oncomine. Lastly, the eight snap-frozen tumorous and adjacent noncancerous adjacent tissues of pancreatic cancer patients used to detect the CDK1 and CEP55 protein levels by western blot.
**Conclusions**. Altogether, the DEGs and hub genes identified in this work can help uncover the molecular mechanisms underlying the tumorigenesis of pancreatic cancer and provide potential targets for the diagnosis and treatment of this disease.

## INTRODUCTION

Pancreatic cancer is one of the most common lethal tumors worldwide, and its overall 5-year survival rate is less than 5% in the United States. PDAC accounts for 95% of all pancreatic cancers (*Siegel, Miller & Jemal, 2018*). Moreover, it has been reported that only 10–15% of pancreatic cancer patients are eligible for tumor resection, a trend that is attributed to the lack of early diagnostic markers and advanced metastasis (*Becker et al., 2016*; *Caruso Bavisotto et al., 2017*). Mounting evidence indicates that abnormal expression and gene variants are related to the tumorigenesis and progression of pancreatic cancer. A study by Caldas C showed that K-ras activation was involved in early events in N-nitroso-bis (2-oxopropyl) amine-induced pancreatic carcinogenesis in hamsters (*Caldas et al., 1994*). In addition, it has been reported that inactivation of SMAD family member 4 (SMAD4) and cyclin dependent kinase inhibitor 2A (CDKN2A) is related to the development of pancreatic cancer (*Pihlak et al., 2018*). Obviously, mutations in tumor suppressor genes are linked to the progression of pancreatic cancer. Undoubtedly, an early diagnosis is beneficial for patients. Therefore, accurate knowledge of the molecular mechanisms involved in the tumorigenesis and proliferation of pancreatic cancer is vital.

Microarray technology and bioinformatic analysis have been extensively applied for screening the expression of genes, miRNAs, lncRNAs, and DNA methylation, helping to identify DEGs and functional pathways relevant to the tumorigenesis and progression of pancreatic cancer. In this work, we aimed to explore the pathogenesis of pancreatic cancer by a computational bioinformatics analysis of gene expression. Three mRNA microarray datasets from the GEO were extracted and analyzed to identify DEGs between PDAC tissues and noncancerous tissues. Subsequently, the functions of these DEGs were evaluated using Gene Ontology (GO) and Kyoto Encyclopedia of Genes and Genomes (KEGG) pathway enrichment analyses. A protein-protein interaction (PPI) network was visualized using Cytoscape, and the Literature Mining Gene Networks module in GenCLiP 2.0 showed that cyclin-dependent kinase 1 (CDK1) has strong interactions with other hub genes. Enrichment analysis of GenCLiP 2.0 suggested that cell division may be involved in the pathogenesis of pancreatic cancer. In the GEPIA database, the mRNA levels of all hub genes were higher in PAAD (pancreatic adenocarcinoma) tumor tissues than in normal tissues. Additionally, survival analysis indicated that abnormal spindle microtubule assembly (ASPM), CDK1, centromere protein F (CENPF), centrosomal protein 55 (CEP55), denticleless E3 ubiquitin protein ligase homolog (DTL), epithelial cell transforming 2 (ECT2), NIMA related kinase 2 (NEK2) and protein regulator of cytokinesis

1 (PRC1) may be associated with the tumorigenesis and development of pancreatic cancer. Overall, 210 DEGs and 13 hub genes that may be candidate biomarkers for pancreatic cancer were identified.

## MATERIAL AND METHODS

### Microarray datasets and data processing

Three human pancreatic cancer mRNA expression datasets (GSE32676 (*Li et al., 2018a*), GSE15471 (*Lu & Li, 2018*) and GSE71989 (*Li et al., 2018a*)) were downloaded from GEO (http://www.ncbi.nlm.nih.gov/geo) (*Edgar, Domrachev & Lash, 2002*), a public functional genomic database containing high-throughput gene expression data, chips and microarrays. They all used the GPL570 [HG-U133_Plus_2] Affymetrix Human Genome U133 Plus 2.0 Array; human PDAC tumors and nonmalignant pancreas samples snap-frozen at the time of surgery were chosen. The GSE32676 dataset contains 25 PDAC tissue samples and seven nonmalignant pancreas samples, GSE15471 contains 39 PDAC tissue samples and 39 nonmalignant pancreas samples, and GSE71989 contains 13 PDAC samples and eight noncancerous samples.

### Identification of DEGs

DEGs between PDAC and nonmalignant samples were screened by GEO2R (http://www.ncbi.nlm.nih.gov/geo/geo2r) (*Shao et al., 2018*), which is an online tool that can be used to compare two or more datasets in a GEO series to identify DEGs according to experimental conditions. Adjusted $P$-values (adj. $P$) and Benjamini and Hochberg false discovery rates were employed as criteria for statistically significant genes and to limit false positives. Probe sets with no corresponding gene symbols or genes with multiple gene probe sets were removed or averaged. Log FC (fold change) $>1$ or $< -1$ and adj. $P < 0.01$ was considered statistically significant. An online tool (http://bioinformatics.psb.ugent.be/webtools/Venn/) was applied to draw Venn diagrams of the DEGs.

### KEGG and GO enrichment analyses of DEGs

The Database for DAVID; (david.ncifcrf.gov/) online tool was used to obtain a synthetic set of functional annotation information for the DEGs (*Huang et al., 2007*; *Le, Ho & Ou, 2018*; *Le, Nguyen & Ou, 2017*; *Meel et al., 2019*). $P < 0.01$ was considered statistically significant.

### PPI network construction and module analysis

A PPI network of the DEGs was established using the STRING (http://string-db.org, version 10.0) database (*Szklarczyk et al., 2015*), and a combination of more than 0.4 was considered statistically significant for the PPI. Subsequently, we visualized the PPI network using Cytoscape, which is an open-source bioinformatics software platform (*Shannon et al., 2003*). Functional module analysis was then performed using MCODE, which is an app for Cytoscape that is used to cluster a given network to a densely connected area based on topology. The standard for selection was set as follows: MCODE scores $>5$, degree cut-off $= 2$, node score cut-off $= 0.2$, max depth $= 100$ and $k$-score $= 2$ (*Li et al., 2017a*).
## Hub gene selection and analysis

Genes with a degree $\geq$10 were chosen as hub genes, and pathways of the hub genes were visualized using ClueGO and CluePedia, which are two plugins of Cytoscape (*Bindea, Galon & Mlecnik, 2013*). Additionally, GenCLiP 2.0 (http://ci.smu.edu.cn) (*Wang et al., 2014*), which facilitates functional annotation and molecular network construction of genes depending on the literature, was used to explore interactions of hub genes. The Gene Cluster with Literature Profiles modules were used to generate statistically overrepresented keywords to annotate genes based on the occurrence of free terms in the literature for a given gene. $P \leq 1 \times 10^6$ and hits $\geq$6 were considered statistically significant (*Li et al., 2017b*). We selected keyword annotation to obtain a cluster analysis heatmap of 13 hub genes. The Literature Mining Gene Networks module was applied to explore the cocitation of hub genes. Furthermore, the expression levels of the 13 hub genes in pancreatic cancer tissues and normal samples were validated using the GEPIA platform, which is a free online database (http://gepia.cancer-pku.cn/) (*Hauptman et al., 2019*). Moreover, overall survival and disease-free survival analyses according to the expression of hub genes were conducted using Kaplan–Meier curve analysis in the cBioPortal (http://www.cbioportal.org) online platform (*Gao et al., 2013*). The relationship between expression level and tumor grade was analyzed using the online database Oncomine (http://www.oncomine.com) (*Rhodes et al., 2004*).

## Transcription factor analysis

The Cytoscape plugin iRegulon (*Janky et al., 2014*) was employed to analyze transcription factors regulating the hub genes. The iRegulon plugin can identify regulons using motifs and track discovery in an existing network or in a set of coregulated genes. Transcription factor information is obtained from databases such as Transfac, Jaspar, Encode, Swissregulon and Homer, which use genome-wide ranking and recovery to detect enriched transcription factor motifs and optimal sets of their direct targets. The cutoff criteria were as follows: enrichment score threshold = 5.0, ROC threshold for AUC calculation = 0.03, rank threshold = 5,000, minimum identity between orthologous genes = 0.05, FDR = 0.001 and the normalized enrichment score (NES) > 10 (*Li et al., 2017b*).

## Patients and tissue specimens

The eight snap-frozen tumorous and adjacent noncancerous adjacent tissues of pancreatic cancer patients used to detect the CDK1 and CEP55 protein levels in this study were provided by the Affiliated Hospital of Nantong University. Our experimental protocols have been subjected to approval by the Institutional Review Board of Affiliated Hospital of Nantong University; all participating patients fully understood the protocols and subscribed informed consent. The cohort of patients included 6 female patients and 2 male patients who underwent surgical resection without chemotherapy or radiotherapy.

## Western blotting

Western blotting was performed as previously described (*Jiao et al., 2019*). After blotting, the membranes were incubated at 4 °C overnight with anti-CDK1 (ab18, diluted 1:1,000, Abcam, MA, USA), anti-CEP55 (ab170414, diluted 1:1000, Abcam, MA), anti-PCNA

(ab29, diluted 1:1,000, Abcam, MA) and anti-GAPDH (#5174; 1:5,000; Cell Signaling Technology, MA, USA) antibodies. This was followed by incubation with anti-mouse IgG (#D110087; 1:2,500; Sangon Biotech, Shanghai, China) or anti-rabbit IgG (#D110058; 1:2,500; Sangon Biotech, Shanghai, China) secondary antibodies. Finally, an enhanced chemiluminescence (ECL) kit was used to visualize the bands, and the Molecular Imager ChemiDoc XRS System (Bio-Rad Laboratories, CA, USA) was used to analyze and quantify the bands.

## Statistical analysis

All data were analyzed using the GraphPad Prism 5.0 software. The results were presented as the mean $\pm$ standard error of mean of at least three independent experiments. The comparison between multiple groups used single-factor analysis of variance, and the comparison of data between separate groups was performed using $t$ test. $P < 0.05$ indicates statistical significance.

# RESULTS

## Identification of DEGs in PDAC

In total, 775, 1793 and 3952 DEGs were identified when comparing PDAC tissue samples and normal tissue samples in the GSE32676, GSE15471 and GSE71989 datasets, respectively. Additionally, 210 (186 upregulated and 24 downregulated) genes were common to all three datasets (Fig. 1A).

## PPI network construction

The PPI network of DEGs was constructed using the STRING database and included 133 genes (125 upregulated and 8 downregulated) with combined scores $>0.4$ (Fig. 1B). As the number of downregulated genes was too small for GO and KEGG enrichment analyses, we only performed this analysis for the upregulated genes.

## GO and KEGG pathway analyses of the PPI network

GO and KEGG pathway analyses were conducted to explore the potential functions and pathways of the upregulated DEGs using DAVID. According to GO analysis, the upregulated DEGs were enriched in cell migration, cell–cell adhesion and cell adhesion biological process (BP) categories (Table 1). The upregulated DEGs were primarily enriched in the extracellular exosome and cytoplasm cell component (CC) categories (Table 1); DEGs were mainly enriched in the cadherin binding involved in cell–cell adhesion and protein homodimerization activity molecular function (MF) categories (Table 1). Additionally, KEGG pathway analysis indicated that the upregulated DEGs were primarily enriched in the ECM-receptor interaction and pathways in cancer (Table 2).

## Hub gene selection and analysis

Thirteen genes were considered to be hub genes with a degree $\geq 10$ (Fig. 1C). Detailed information about these hub genes is presented in Table 3. The pathways of the hub genes were visualized using ClueGO and CluePedia, which are two plugins of Cytoscape (Fig. 2). Subsequently, literature mining was performed in GenCLiP 2.0 to explore hub

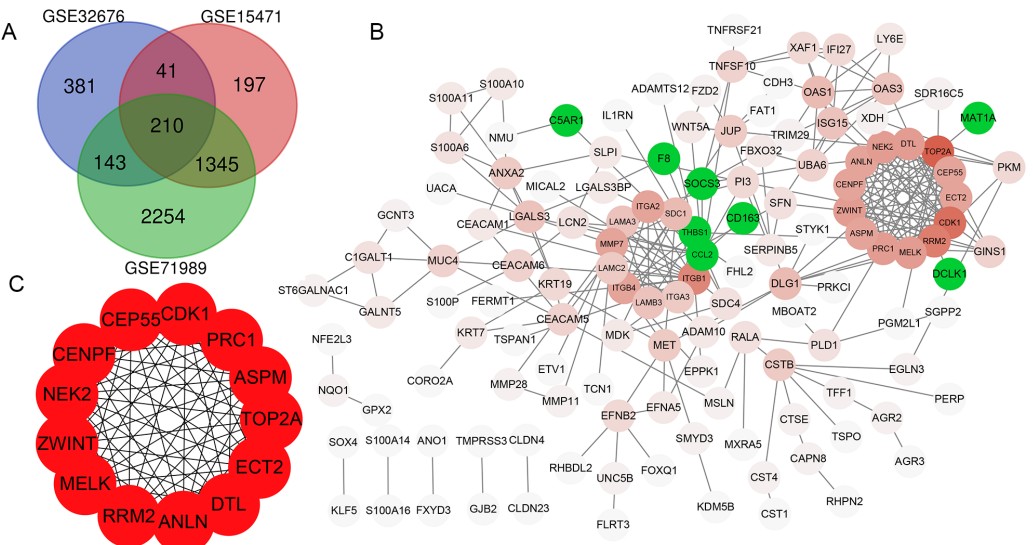

**Figure 1** **Venn diagram, PPI network and the most significant modules of DEG.** (A) Venn diagram analyzing the DEGs of GSE32676, GSE15471 and GSE71989 datasets, and an overlap of 210 genes was identified. (B) The PPI network of DEGs was constructed by Cytoscape. Upregulated genes are marked in red, and the depth of color represents the gene interaction degree with other genes; downregulated genes are marked in green. (C) Interaction network of 13 hub genes.

gene interactions. The cocitation network of these 13 hub genes in the published literature is displayed in Fig. 3A. The results showed that 5 genes, DNA topoisomerase II alpha (TOP2A), PRC1, ECT2, RRM2 and CEP55 interact with CDK1. TOP2A was the top gene because it has been mentioned in 23 published literature sources as interacting with CDK1. The detailed results of previous studies on these genes are shown in Table 4. Figure 3B illustrates the significant results from the enrichment analysis of the 13 hub genes, with cell division being the most related biological function at 76.9% enrichment.

## Transcription factor analysis of hub genes
Transcription factor analysis of the 13 hub genes was conducted using iRegulon, a Cytoscape plugin, and a normalized enrichment score (NES) > 10 was considered to be significant. The transcriptional regulation network of these hub genes is shown in Fig. 3C. The transcription factors with NES > 10 were NFYC (NES = 15.131, targets = 11), TFDP3 (NES = 11.922, targets = 11), NFYA (NES = 11.193, targets = 9), and E2F4 (NES = 10.426, targets = 13).

## mRNA expression levels of the 13 hub genes in PAAD
To confirm the expression levels of the 13 identified hub genes, related published data were obtained from TCGA datasets and analyzed using the GEPIA platform. As expected, the results showed that all hub genes were more highly expressed in tumor tissues than in normal tissue samples (Fig. 4).

## Survival analyses of hub genes
Kaplan–Meier curve analysis was used to analyze correlations between overall survival and the hub genes. PDAC patients with alterations in anillin actin-binding protein (ANLN),
**Table 1  GO analysis of upregulated DEGs in pancreatic ductal adenocarcinoma.**

| GO ID | Description | Gene count | *P*-value |
|---|---|---|---|
| GO-BP Terms | | | |
| GO:0016477 | Cell migration | 13 | 1.83E−07 |
| GO:0098609 | Cell–cell adhesion | 13 | 2.08E−05 |
| GO:0007160 | Cell–matrix adhesion | 8 | 3.55E−05 |
| GO:0022617 | Extracellular matrix disassembly | 7 | 1.20E−04 |
| GO:0048333 | Mesodermal cell differentiation | 4 | 1.58E−04 |
| GO:0031581 | Hemidesmosome assembly | 4 | 2.10E−04 |
| GO:0090004 | Positive regulation of establishment of protein localization to plasma membrane | 5 | 2.26E−04 |
| GO:0007155 | Cell adhesion | 15 | 2.45E−04 |
| GO:0007229 | Integrin-mediated signaling pathway | 7 | 5.08E−04 |
| GO-CC Terms | | | |
| GO:0070062 | Extracellular exosome | 63 | 1.50E−10 |
| GO:0005925 | Focal adhesion | 16 | 7.65E−06 |
| GO:0005913 | Cell–cell adherens junction | 14 | 1.91E−05 |
| GO:0005615 | Extracellular space | 28 | 3.05E−04 |
| GO:0005886 | Plasma membrane | 61 | 5.23E−04 |
| GO:0005737 | Cytoplasm | 72 | 9.52E−04 |
| GO:0009986 | Cell surface | 15 | 9.53E−04 |
| GO-MF Terms | | | |
| GO:0098641 | Cadherin binding involved in cell–cell adhesion | 13 | 3.05E−05 |
| GO:0042803 | Protein homodimerization activity | 18 | 8.70E−04 |
| GO:0005509 | Calcium ion binding | 16 | 0.004808 |
| GO:0005515 | Protein binding | 102 | 0.013495 |

**Notes.**
GO, gene ontology; BP, biological process; CC, cellular component; MF, molecular function.

**Table 2  KEGG pathway enrichment analysis of upregulated DEGs in pancreatic ductal adenocarcinoma.**

| ID | Description | Gene count | *P*-value |
|---|---|---|---|
| hsa04512 | ECM-receptor interaction | 9 | 1.82E−06 |
| hsa05200 | Pathways in cancer | 12 | 0.001424 |
| hsa05205 | Proteoglycans in cancer | 8 | 0.003332 |
| hsa04510 | Focal adhesion | 8 | 0.003924 |
| hsa04115 | p53 signaling pathway | 5 | 0.004128 |
| hsa04151 | PI3K-Akt signaling pathway | 9 | 0.019238 |

**Notes.**
KEGG, Kyoto Encyclopedia of Genes and Genomes.

ASPM, CDK1, CENPF, CEP55, DTL, ETC2, NEK2, TOP2A and PRC1 exhibited poor overall survival (Fig. 5). PDAC patients with ASPM, CDK1, CENPF, CEP55, DTL, ETC2, NEK2, PRC1, ribonucleotide reductase regulatory subunit M2 (RRM2), TOP2A, ZW10

**Table 3  Description of 13 hub genes of pancreatic ductal adenocarcinoma.**

| No. | Gene symbol | Full name | Function |
|---|---|---|---|
| 1 | TOP2A | DNA topoisomerase II alpha | Controlling and altering the topologic states of DNA during transcription. Involving in chromosome condensation, chromatid separation, and the relief of torsional stress. |
| 2 | CDK1 | Cyclin dependent kinase 1 | Essential for G1/S and G2/M phase transitions of eukaryotic cell cycle and mitotic cyclins. |
| 3 | RRM2 | Ribonucleotide reductase regulatory subunit M2 | Catalyzing the formation of deoxyribonucleotides from ribonucleotides. |
| 4 | PRC1 | Protein regulator of cytokinesis 1 | A substrate of several CDKs and necessary for polarizing parallel microtubules and concentrating the factors responsible for contractile ring assembly. |
| 5 | NEK2 | NIMA related kinase 2 | A serine/threonine-protein kinase that is involved in mitotic regulation. |
| 6 | ZWINT | ZW10 interacting kinetochore protein | Clearly involved in kinetochore function. |
| 7 | DTL | Denticleless E3 ubiquitin protein ligase homolog | Cellular response to DNA damage stimulus. |
| 8 | MELK | Maternal embryonic leucine zipper kinase | G2/M transition of mitotic cell cycle apoptotic process. |
| 9 | CENPF | Centromere protein F | Associating with the centromere-kinetochore complex and playing a role in chromosome segregation during mitotis. |
| 10 | CEP55 | Centrosomal protein 55 | Cranial skeletal system development and establishment of protein localization. |
| 11 | ANLN | Anillin actin binding protein | An actin-binding protein that plays a role in cell growth and migration, and in cytokinesis. |
| 12 | ASPM | Abnormal spindle microtubule assembly | Essential for normal mitotic spindle function in embryonic neuroblasts and regulating neurogenesis. |
| 13 | ECT2 | Epithelial cell transforming 2 | A guanine nucleotide exchange factor and transforming protein that is related to Rho-specific exchange factors and yeast cell cycle regulators. |

**Notes.**

CDKs, cyclin-dependent kinases.

interacting kinetochore protein (ZWINT), and ANLN alterations exhibited poor disease-free survival (Fig. 6). Among these genes, we selected CDK1 and CEP55 for further study with regard to overall survival and disease-free survival times.

## Oncomine analysis of CDK1 and CEP55 in cancer vs. normal tissue

Oncomine analysis of cancer vs. normal tissue indicated that CDK1 and CEP55 were significantly overexpressed in pancreatic cancer in different datasets (Figs. 7A and 7B). In the Grutzmann Pancreas dataset, CDK1 and CEP55 mRNA expression was higher in pancreatic cancer tissues than in normal pancreatic tissues (Figs. 7C and 7D). Additionally, higher mRNA levels of CDK1 and CEP55 were associated with tumor grade (Figs. 7E and 7F).

## The expression of CDK1 and CEP55 in clinical specimens

To validate the above results, we detected CDK1 and CEP55 protein levels in clinical specimens. In the clinical specimens, CDK1 (Figs. 8A and 8C) and CEP55 (Figs. 8B and 8D) protein levels were significantly elevated in pancreatic cancer tissue samples compared with adjacent nontumor tissues.
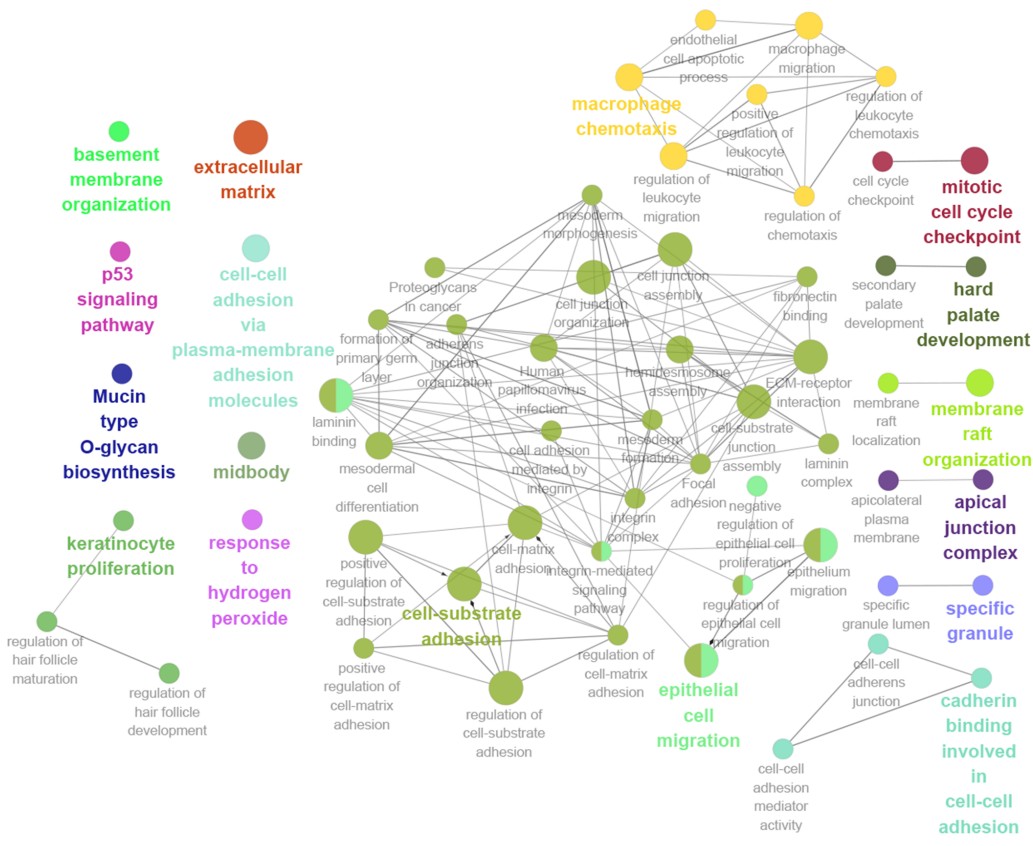

**Figure 2  GO and KEGG pathway analysis of hub genes using ClueGO and CluePedia.**

# DISCUSSION

In recent years, microarray technology has been extensively applied to reveal genetic alterations in tumors. Therefore, microarray analysis is a tool for revealing biomarkers for the diagnosis, treatment, and prognosis of pancreatic cancer. In this study, our results demonstrated that the upregulated DEGs were obviously enriched in the BP categories cell migration and cell–cell adhesion. Many studies have shown that migration and invasion are basic characteristics of pancreatic cancer (*Zhuo et al., 2018*). Moreover, cell–cell adhesion might be involved in cell stretching and movement, which are the molecular bases of the important physiological and pathological processes of tumor invasion and metastasis (*Nobes & Hall, 1999*; *Serrill, Sander & Shih, 2018*). For the CC category, the upregulated DEGs were primarily enriched in extracellular exosomes, which are nanoscale membrane vesicles with diameters ranging from 40 to 100 nm. A growing number of studies have shown that tumor-derived exosomes are associated with tumor development, metastasis, and drug resistance mechanisms (*Jiao et al., 2018*). Furthermore, in the MF category, upregulated DEGs were mainly enriched in cadherin binding involved in cell–cell adhesion and protein homodimerization activity, which are associated with invasion and metastasis. According to our KEGG pathway enrichment results, the upregulated DEGs were primarily

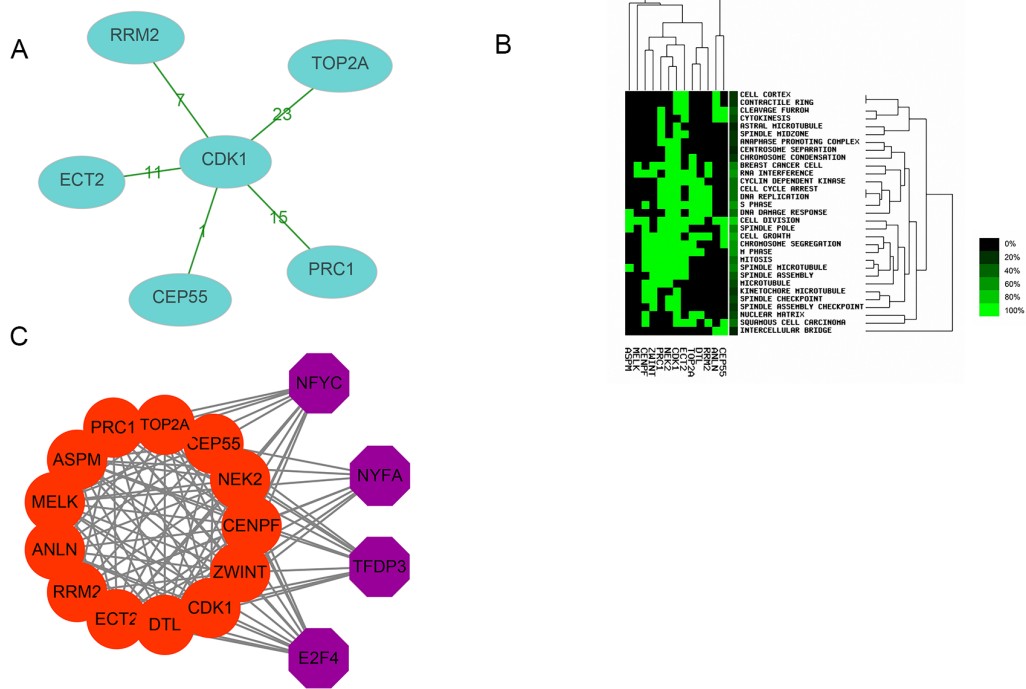

**Figure 3** **Interactions, enrichment and transcription factor analysis of 13 hubgenes.** (A) The cocitation network of 13 hub genes; the number on the line shows the number of studies cocited. (B) Heatmap of enrichment analysis of 13 hub genes; the depth of green color represents the degree of enrichment. (C) Transcription factor analysis of hub genes; the red nodes represent hub genes, and the purple nodes represent transcription factors.

**Table 4** **Hub genes identified by the present study using Genclip 2.0.**

| Gene | Co-genes | Co-cite | Total |
|------|----------|---------|-------|
| CDK1 | 5 | 53 | 6,822 |
| TOP2A | 1 | 23 | 6,370 |
| RRM2 | 1 | 7 | 395 |
| CEP55 | 1 | 1 | 58 |
| PRC1 | 1 | 15 | 334 |
| ECT2 | 1 | 11 | 171 |

enriched in the ECM-receptor interaction. During the development of tumors, the ECM experiences a remodeling process that is similar to the process of embryonic development. The most important feature of this remodeling is the change in the molecular composition of the ECM, whereby the reconstituted ECM creates a loose microenvironment for the proliferation and differentiation of tumor cells, leading to high rates of proliferation, poor differentiation, invasion and metastasis (*Jin & Liu, 2018*).

In our study, we showed that TOP2A, PRC1, ECT2, RRM2 and CEP55 can interact with CDK1 and TOP2A is the gene most closely related to CDK1. Previous studies have provided evidence about the expression of CDK1 and TOP2A in pancreatic cancer, but the

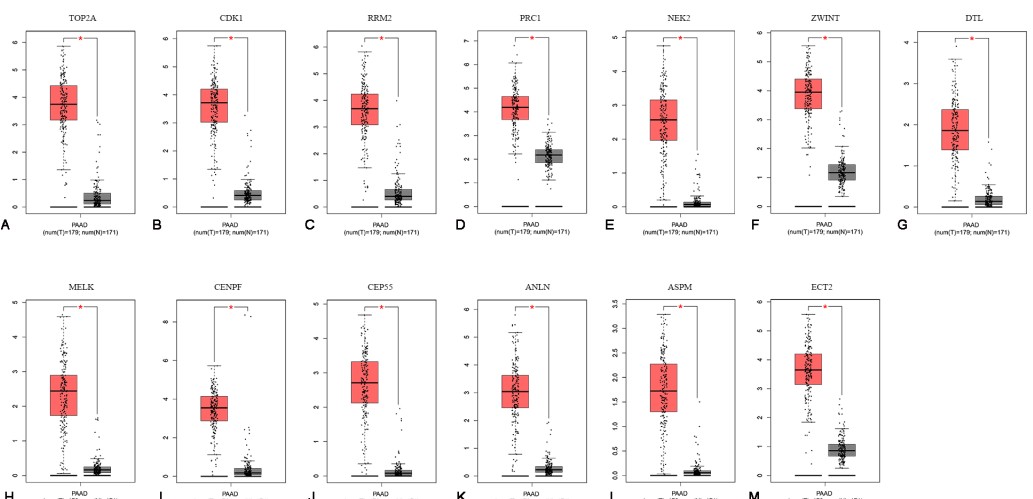

**Figure 4  The mRNA expression levels of 13 hubgenes in PAAD.** The published online data of gene mRNA expression level were analyzed by GEPIA platform. These 13 hubgenes were all higher than in PAAD tissues, compared with those in normal tissues, including TOP2A, CDK1, RRM2, PRC1, NEK2, ZWINT, DTL, MELK, CENPF, CEP55, ANLN, ASPM, ECT2. *$P < 0.05$.

correlation between them in pancreatic cancer has not been fully elucidated (*Kalimutho et al., 2018*; *Li et al., 2017a*; *Shi et al., 2015*; *Xu et al., 2016*). Cyclin-dependent kinases (CDKs) are important driving factors of the human cell cycle (*Zhao et al., 2013*). Topoisomerases, including topoisomerase 1 (TOP1) and topoisomerase 2 (TOP2), are key ribozymes that mainly participate in cell growth by breaking and reconnecting DNA strands to change DNA topology (*Liu et al., 2018*). We predict that the cyclin A2-CDK1-TOP2A axis plays an important role in tumor development. Furthermore, CDK1 might phosphorylate TOP2A to promote S phase transition and influence the progression of pancreatic cancer, but follow-up research is needed to determine the specific mechanism of the interaction (*Kalimutho et al., 2018*). Our enrichment analysis of 13 hub genes further verified the relationship between metastasis and the malignant progression of pancreatic cancer.

The transcription factors NFYC, TFDP3, POLE3 and E2F4 are closely linked to hub genes in pancreatic cancer. NFYC is a histone-fold domain-containing transcription factor engaged in chromatin remodeling, establishing permissive chromatin modifications at CCAAT motifs in promoters (*Bieniossek et al., 2013*). Deletion of NFYC halts cell cycle progression, predominantly by causing G2/M arrest, and concurrent gain of NFYC may serve to model an aberrant epigenome that promotes a proliferative and relatively undifferentiated state (*Benatti et al., 2011*). Most E2Fs are localized in the nucleus, but E2F4 shows cell cycle-specific localization and can be found in the nucleus of cycling cells in G0, early G1 and G2 phases (*Lindeman et al., 1997*). When present in the nucleus, E2F4 has functions necessary for the induction of mitosis; interestingly, E2F4 also appears to have a role in the cytoplasm during multiciliogenesis (*Van Amerongen et al., 2010*). TFDP3 is expressed in most cancer tissues and potentially plays a role in cell differentiation and proliferation. TFDP3 is a novel negative regulator of E2F that can enhance both the DNA

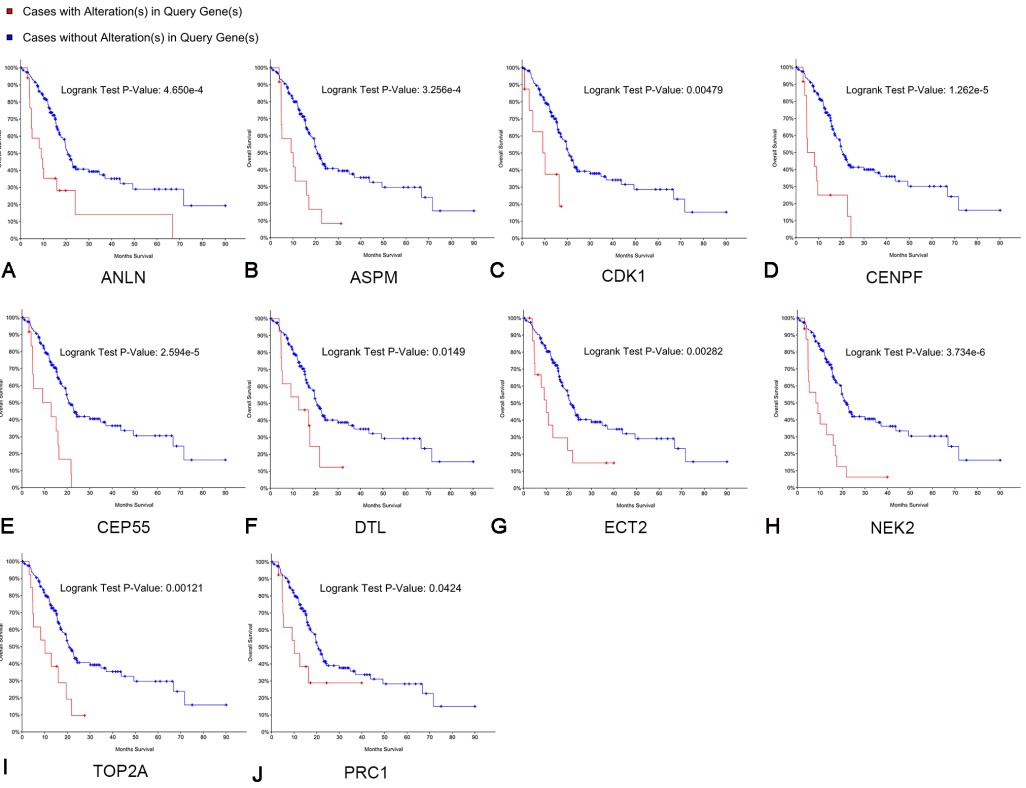

**Figure 5  Overall survival analyses of hub genes performed using the cBioPortal online platform.** A log rank test $P < 0.05$ was considered statistically significant.

binding and transcriptional activity of E2F through the formation of heterodimers; it also potentially plays an important role in the process of tumor development independent of pRb (*Ma et al., 2014*). POLE3 is similar to the first subclass of core histones with respect to regulation, and POLE3 expression is upregulated at the onset of S phase. E2F4 is often associated with promoters in G0, is minimally bound to the POLE3 gene in starved cells and maximally in cells expressing POLE3 at high levels. These findings indicate that these transcription factors may regulate the progression of pancreatic cancer, which provides direction for our future research (*Bolognese et al., 2006*).

Expression analysis of PAAD tissues and normal tissues based on the TCGA database also indicated higher levels of TOP2A (*Jiao et al., 2019*), CDK1 (*Jing et al., 2019*), RRM2 (*Zhao et al., 2019*), PRC1 (*Mao et al., 2019*), NEK2 (*Deng et al., 2019*), ZWINT (*Obuse et al., 2004*), DTL(*Cui et al., 2019*), MELK (*Meel et al., 2019*), CENPF (*Chen et al., 2019*), CEP55 (*Hauptman et al., 2019*), ANLN (*Wang et al., 2019*), ASPM (*Hsu et al., 2019*), and ECT2 (*Daulat et al., 2019*) in tumor tissues. Previous studies have provided abundant evidence about the function of the thirteen identified hub genes in cancer development. Mutations in ANLN, ASPM, CDK1, CENPF, CEP55, DTL, ETC2, NEK2, TOP2A and PRC1 affect overall survival and disease-free survival in pancreatic cancer, whereas mutations in RRM2 and ZWINT affect disease-free survival. Hence, we reasoned that ANLN, ASPM, CDK1, CENPF,

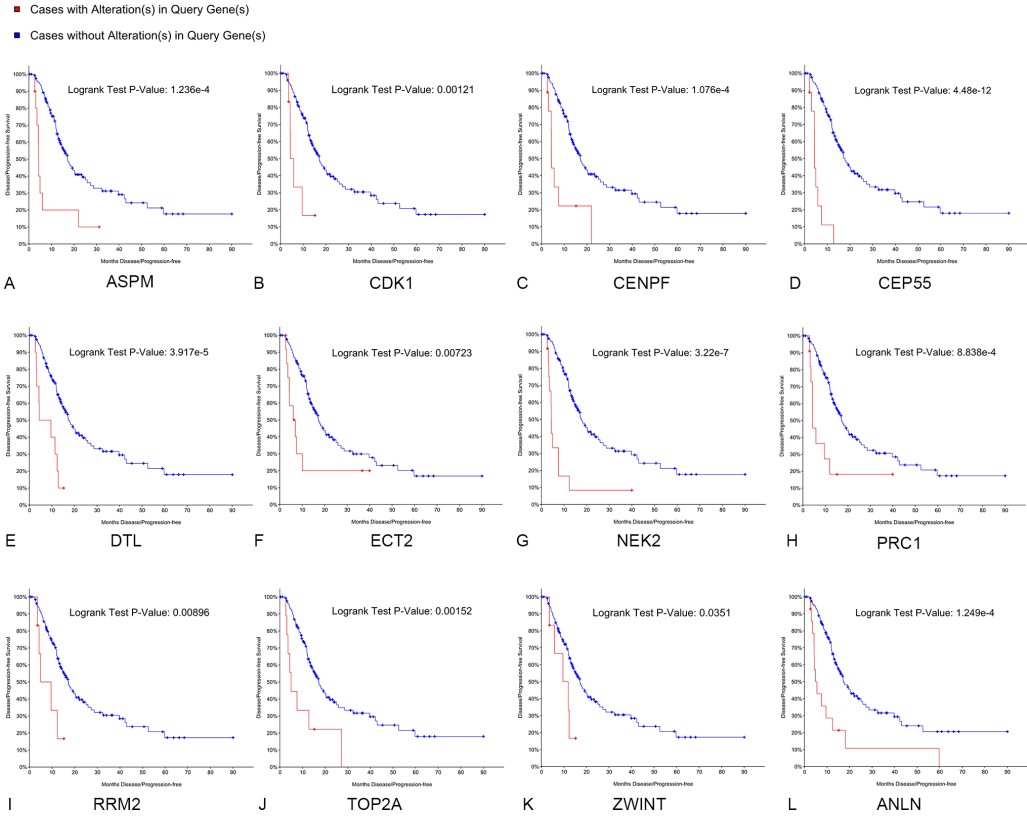

**Figure 6 Disease-free survival analyses of hubgenes (A–L) were performed using the cBioPortal online platform.** A log rank test $P < 0.05$ was considered statistically significant.

CEP55, DTL, ETC2, NEK2, TOP2A and PRC1 might be preferable prognostic factors that are positively related to pancreatic cancer. Additionally, *Lu, Li & Liao (2019)* analyzed the microarray datasets GSE32676, GSE15471, GSE71989 and GSE19650 to identify five upregulated hub genes, including MELK, MET, THBS1, TOP2A and SDC1; *Li et al. (2018a)* analyzed the microarray datasets GSE71989, GSE15471, GSE16515, GSE32676, GSE41368 and GSE28735 and found that five genes (BIRC5, CKS2, ITGA3, ITGA6 and RALA) were significantly associated with survival time in patients with pancreatic duct adenocarcinoma; *Zhu et al. (2017)* studied the five GEO datasets (GSE15471, GSE16515, GSE18670, GSE32676, GSE71989) and reported that GJB2 and ERO1LB dysregulation was associated with tumorigenesis in pancreatic adenocarcinoma. Within addition to these hub genes, we supplied some new hub genes that may be associated with the tumorigenesis and development of pancreatic cancer.

Our results showed that among these hub genes, TOP2A is the most closely related gene to CDK1, TOP2A and CDK1, and it has been previously reported as a biomarker for pancreatic cancer (*Kokkinakis, Liu & Neuner, 2005*); however, to date, little attention has been paid to CEP55 and its possible relationship with CDK1. Indeed, only one study reported that CEP55 can inhibit CDK1 phosphorylation and proteolysis mediated by the anaphase-promoting complex to induce anaphase I in oocytes (*Zhou et al., 2019*).

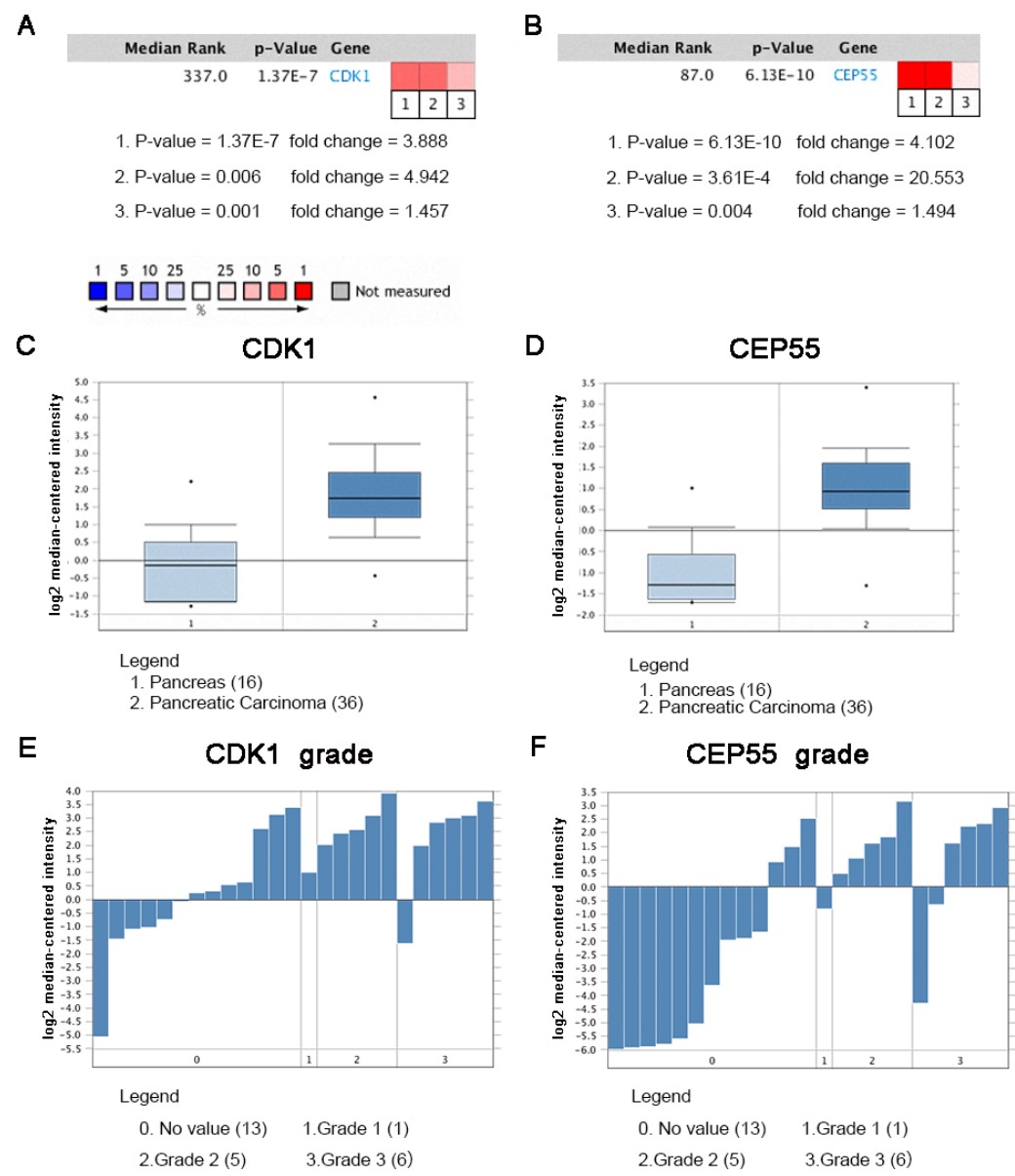

**Figure 7** **Oncomine analysis of CDK1 and CEP55 in cancer vs. normal tissue.** Heat maps of CDK1 and CEP55 gene expression in clinical pancreatic cancer samples vs. normal tissues. 1. Pancreatic Ductal Adenocarcinoma Epithelia vs. Normal Grutzmann Pancreas, Neoplasia, 2004. 2. Pancreatic Carcinoma vs. Normal Pei Pancreas, Cancer Cell, 2009. 3. Pancreatic Carcinoma vs. Normal Segara Pancreas, Clin Cancer Res, 2005. (C) CDK1 mRNA expression and (D) CEP55 mRNA expression in pancreatic cancer compared with normal pancreatic tissues in the Grutzmann Pancreas dataset. Association between the expression of (E) CDK1 and (F) CEP55 and tumor grade in the Grutzmann Pancreas dataset. $P < 0.05$ was considered statistically significant.

Nonetheless, the relationship between CEP55 and CDK1 and the mechanism involved have not been studied in pancreatic cancer cells. It is well known that pancreatic cancer has a high degree of malignancy and a short disease course, which is largely attributed to the fact that it readily metastasizes to and invades adjacent organs. CEP55 belongs to the

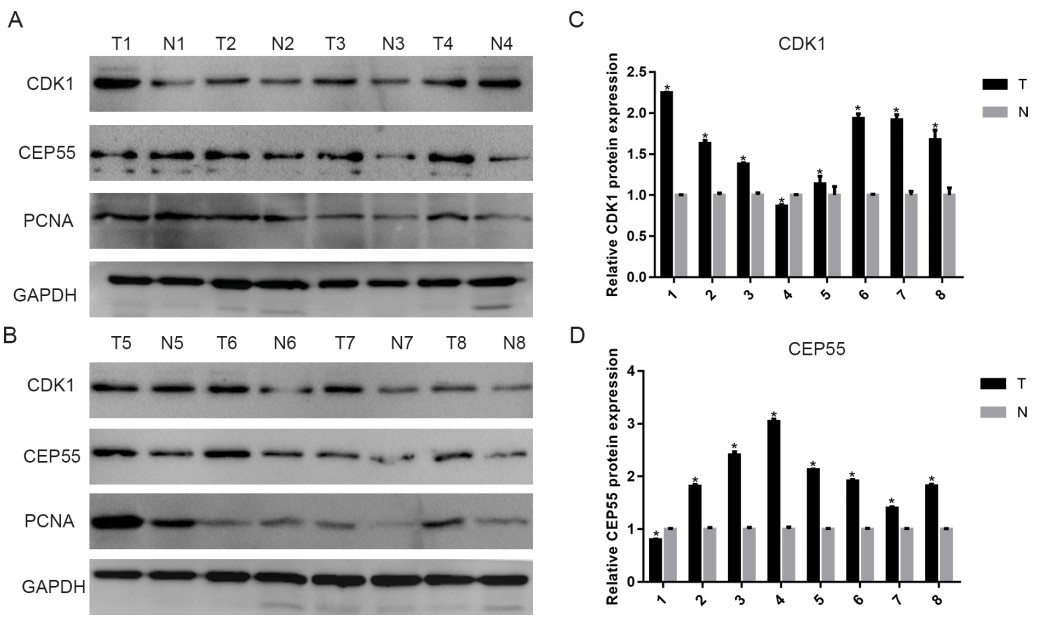

**Figure 8** **The expression of CDK1 and CEP55 in clinical specimens.** (A, B) Western blotting detection of CDK1 and CEP55 protein in eight snap-frozen tumorous and non-cancerous adjacent tissues of pancreatic cancer patients. (C, D) The bar chart reveled that the expression ratio of CDK1 and CEP55 to GAPDH by densitometry. Data are represented as the mean ± SEM (*$P < 0.05$, tumor tissues compared with the non-cancerous adjacent tissues).

centrosomal family of proteins, which plays an important role in critical cell functions. Mounting evidence shows that CEP55 is highly expressed in multiple cancers, such as colon cancer, hepatocellular carcinoma and bladder cancer (*Gao & Wang, 2015*; *Li et al., 2018b*; *Singh et al., 2015*). Moreover, overexpression of CEP55 accelerates the cell cycle transition in gastric cancer, and low expression of CEP55 inhibits cell growth in breast cancer and gastric cancer (*Kalimutho et al., 2018*; *Tao et al., 2014*). These results demonstrate that CEP55 may serve an oncogenic role and a potential target for tumor treatment. CEP55 can also promote aggressive behavior in pancreatic cancer cells by activating the NF-$\kappa$B pathway (*Peng et al., 2017*). Hence, the underlying mechanism of the proinvasive and prometastatic effects of CEP55 in pancreatic cancer needs to be further studied. Last, we preliminarily validated the ChIP analyses by detecting CDK1 and CEP55 protein levels in eight snap-frozen tumorous and adjacent noncancerous adjacent tissues of pancreatic cancer patients. The results showed that CDK1 and CEP55 were significantly overexpressed in pancreatic tumorous tissues compared to normal tissues, which is consistent with the results in different datasets. However, our sample size was relatively small, so we could not verify the link between clinical samples and prognosis. Fortunately, *Piao et al., (2019)* validated that high expression of CDK1 was correlated with the short survival of pancreatic cancer patients by analyzing 99 cases of surgically resected pancreatic cancer samples and 71 cases of normal pancreases. *Peng et al. (2017)* reported that CEP55 expression was an independent prognostic factor of patient outcome and that CEP55 protein expression levels in pancreatic cancer specimens were inversely correlated with survival time by analyzing

126 archived paraffin-embedded pancreatic cancer specimens with immunohistochemical staining using an antibody against human CEP55. The results were in accordance with the Kaplan–Meier curve analysis online.

## CONCLUSION

In summary, by analyzing multiple datasets from the GEO database and validating the results with the TCGA and Oncomine databases, our present work identifies dominant genes, their interaction network and possible transcription factors involved during the progression and metastasis of pancreatic cancer. Some relationships between hub genes and transcription factors have never been reported to influence the progression of pancreatic cancer and may serve as potential targets for pancreatic cancer therapy. However, due to the low quantity of gene probes in our selected datasets, the number of discovered DEGs was strikingly limited. More genes and noncoding RNAs should be detected to enrich the network for a more comprehensive and integrated understanding of pancreatic cancer development. In addition, we did not analyze the expression of hub genes, such as CDK1, in pancreatic patients with or without lymphatic metastasis or use ROC analysis to explain the prognosis and diagnostic value of the genes. We will collect these data in our future research to complete our analysis of the prognosis and diagnostic value of these genes.

### Funding

This study was supported by grants from the Natural Science Foundation of China (grant No. 81472272 and 81602114), the Key Research and Development Plan of Jiangsu Province (no. BE2019692), the Postdoctoral Science Foundation of China (grant No. 2017M620221), the Social Development Foundation of Nantong City (grant no. MS22018006, MS12019018, MS12019020, JC2019032), and the Teaching Research Project of Affiliated Hospital of Nantong University (Tfj 18006). The funders had no role in study design, data collection and analysis, decision to publish, or preparation of the manuscript.

### Grant Disclosures

The following grant information was disclosed by the authors:
Natural Science Foundation of China: 81472272, 81602114.
Key Research and Development Plan of Jiangsu Province: BE2019692.
Postdoctoral Science Foundation of China: 2017M620221.
Social Development Foundation of Nantong City: MS22018006, MS12019018, MS12019020, JC2019032.
Teaching Research Project of Affiliated Hospital of Nantong University: Tfj 18006.

### Competing Interests

The authors declare there are no competing interests.

## Author Contributions

- Dandan Jin conceived and designed the experiments, analyzed the data, prepared figures and/or tables, and approved the final draft.
- Yujie Jiao, Jie Ji, Wei Jiang, Wenkai Ni and Yingcheng Wu performed the experiments, prepared figures and/or tables, and approved the final draft.
- Runzhou Ni, Cuihua Lu, Hongbing Ni and Jinxia Liu analyzed the data, authored or reviewed drafts of the paper, and approved the final draft.
- Lishuai Qu analyzed the data, prepared figures and/or tables, authored or reviewed drafts of the paper, and approved the final draft.
- Weisong Xu and MingBing Xiao conceived and designed the experiments, authored or reviewed drafts of the paper, and approved the final draft.

## Human Ethics

The following information was supplied relating to ethical approvals (i.e., approving body and any reference numbers):

The Institutional Review Board of Affiliated Hospital of Nantong University approved the study.

## Data Availability

The microarray datasets are available at Gene Expression Omnibus (GEO): GSE32676, GSE15471 and GSE71989.

## Supplemental Information

Supplemental information for this article can be found online at http://dx.doi.org/10.7717/peerj.9301#supplemental-information.

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
