# Peer review of "Identification of prognostic risk factors for pancreatic cancer using bioinformatics analysis"

_PeerJ, doi:10.7717/peerj.9301_

## Round 0.1 · original submission · Major Revisions

Dear Dr. MingBing Xiao,

Thank you for submitting your manuscript to PeerJ. Your manuscript has been assessed by three reviewers and myself as an Academic Editor. Reviewers have expressed interest in the study, but also have mentioned several criticisms and suggestions regarding the findings of the study. In particular, the manuscript lacks explanation to choose different criteria for insilico analysis. In addition to further analysis as suggested by reviewers, the study requires thorough discussion to justify the importance of the present findings to identify prognostic risk factors for pancreatic cancer. You are also asked to improve the manuscript by removing spelling mistakes and grammatical errors throughout the paper. I am including reviewer comments, which I hope you will find useful and constructive. If it is possible to address the concerns raised with additional data and/or discussion, we would be interested in considering a revised version of the manuscript. In this case, I would strongly encourage you to pay close attention to the interpretation and the discussion of the data, in light of the reviewer’s feedback. While preparing your revised manuscript, I suggest you submit a detailed response to the reviewer’s comment along with the revised version of the manuscript.

·

Basic reporting

no comment

Experimental design

no comment

Validity of the findings

no comment

Additional comments

In this study, the authors identified DEGs in Pancreatic Cancer, and demonstrated that DEGs were mainly involved in cell migration, cell-cell adhesion and cell adhesion; the cellular components were enriched in extracellular exosome and cytoplasm; and the molecular functions were enriched in cadherin binding involved in cell-cell adhesion and protein homodimerization activity. Moreover, all hubgenes had higher levels in PAAD (Pancreatic adenocarcinoma) tumor tissues than that in normal samples, and the relationship between prognosis and prognosis was further analyzed. These results are consistent to previous study which showed stability of cell structure and cell adhesion is a major factor in the formation of pancreatic cancer. Examination design was a well-thought-out, and acquired results were consistent. The manuscript appears seems to be solid, but I found problems in Discussion.
Specific concern:
1. Discussion section should summarize the potential underlying mechanism and discuss the present results rather than giving a review of previous findings.
2. There are still many problems in the language of the article, which require further modification and improvement.

·

Basic reporting

I commend the authors for their extensive data set. In addition, the manuscript is clearly written in profession, unambiguous language. And your research question well defined, relevant and meaningful.

Experimental design

Methods described with sufficient detail and information to replicate. However, there are some weaknesses

Validity of the findings

1.According to the discussion of your manuscript, TOP2A is more related to CDK1, but why CEP55 was chosen for further research?
2.Your statistical analysis needs be consummated. I suggest that you analyze the expression of hubgenes, such as CDK1, in pancreatic patients with or without lymphatic metastasis and add ROC to explain the prognosis and diagnostic value of the genes.
3.The relationship between CDK1and CEP55 with the tumor grade is not right in the Oncomine。

Reviewer 3 ·

Basic reporting

- Resolution of figures needs to be improved. It is not easy to see clearly.
- There are minor grammatical errors & typos that can be improved in the revised version.
- To mention Gene Ontology, the authors should provide more relevant bioinformatics studies that worked on it, such as https://doi.org/10.1016/j.ab.2018.06.011, https://doi.org/10.1016/j.jmgm.2017.01.003, and https://doi.org/10.1016/j.ab.2019.03.017.

Experimental design

- The authors have to provide more information to show the literate reviews as well as how they fill an identified knowledge gap in this problem. Recently, this lacks much information in the current version.
- What is the idea of using different levels of p-value in their methods? I checked that sometimes the authors used p<0.05 as significant level, but sometimes is 1x10^6 or 0.01. I think it is not consistent and there must have some explanations on their choices. Without them, it is not easy to say that their results are significant enough.
- How to select a set of parameters in Cytoscape?
- Similarly, how did the authors choose the optimal cutoff criteria in transcription factor analysis?

Validity of the findings

- A big missing is that the authors indeed didn't link their found genes with pancreatic cancer. How do the functions of these genes play in pancreatic cancer? It is interesting and they need discussing more of their findings.

Additional comments

No comment

---

## Round 0.2 · Minor Revisions

Dear MingBing Xiao, 

Thank you for providing the revised manuscript. The revised manuscript has been assessed by three reviewers and myself as an AcademicEditor. The revised version has certainly been improved. The language of the article, as well as the flow of the writing, has significantly improved. However, one of the three reviewers has expressed criticisms as well as suggestions to include more specific details regarding the material/tissues/ChIP/method used for the study. For details, I am including reviewer comments, which I hope you will find useful and constructive. 

I would strongly encourage you to pay close attention to the interpretation and the discussion of the data, in light of the reviewers' feedback. While preparing your revised manuscript, Please submit a detailed response to the reviewer’s comment along with the revised version of the manuscript. These suggestions are essential to incorporate before publication. Finally, I encourage authors to submit a revised version of the manuscript to be considered for final publication in PeerJ. 

Looking forward to the revised manuscript.

Best wishes,
Shalu Jhanwar

·

Basic reporting

no comment

Experimental design

no comment

Validity of the findings

no comment

Additional comments

In this study, the authors found that 210 differentially expressed genes (DEGs) were identified, including 186 upregulated and 24 downregulated geneswas in pancreatic cancer. the DEGs and hub genes identified in this work can help uncover the molecular mechanisms underlying the tumorigenesis of pancreatic cancer and provide potential targets for the diagnosis and treatment of this disease. Examination design was a well-thought-out, and acquired results were consistent. The manuscript appears seems to be solid. However, there are some fatal problems.
Specific concern:
1. There have been articles reporting similar research content by authors. (Lu W, et al. dentification of Key Genes and Pathways in Pancreatic Cancer Gene Expression Profile by Integrative Analysis; Zhou J, et al. Identification of novel genes associated with a poor prognosis in pancreatic ductal adenocarcinoma via a bioinformatics analysis). It greatly reduced the novelty of the work.
2. Please inform the author of the specific information of the chip (chip type, source, etc.)
3. It is disturbing that the authors concealed the fact that this is a re-analysis study. Why did the authors re-analyse data generated by others and then did not even mentioned the original study?
4. This article only carries on the biological analysis, has not carried on the clinical sample and the basic experiment confirmation, the experimental result's authenticity and the reliability are doubtful.
5. In this paper, the authors should make clear which type of pancreatic cancer gene chip to study.
6. There are still many problems in the language of the article, which require further modification and improvement.

Reviewer 3 ·

Basic reporting

No comment.

Experimental design

No comment.

Validity of the findings

No comment.

Additional comments

My previous comments have been addressed satisfactorily.

Reviewer 4 ·

Basic reporting

Satisfactory

Experimental design

Experiments are well conducted.

Validity of the findings

Findings are valid as GEO2R software was used and its well know for finding DEG.

Additional comments

Authors have made substantial corrections to the ms.

---

## Round 0.3 · Major Revisions

Dear Dr. Xiao,

Thank you for providing the revised manuscript. The revised manuscript has been assessed by a reviewer and myself as an Academic Editor. The revised version has certainly been improved. The language of the article, as well as the flow of the writing, has significantly improved. However, the reviewer has expressed specific concerns in the present study. More specifically, it is suggested to analyze/discuss/comment on a comparison of the outcome of the present study with the previously published reports. For details, I am including reviewer comments, which I hope you will find useful and constructive. I would strongly encourage you to pay close attention to the interpretation and the discussion of the data, in light of the reviewers' feedback. While preparing your revised manuscript, I suggest you submit a detailed response to the reviewer’s comment along with the revised version of the manuscript. These suggestions are essential to incorporate before publication. Finally, I encourage authors to submit a revised version of the manuscript to be considered for final publication in PeerJ.

I am looking forward to the revised manuscript.

Best wishes,
Shalu Jhanwar

·

Basic reporting

The innovation of the article is general, the author's work is not bad, but there are still some problems in the article

Experimental design

good

Validity of the findings

goog

Additional comments

The innovation of the article is general, the author's work is not bad, but there are still some problems in the article
1. The author should collect some clinical samples to verify the result.
2. The language of the article needs to be further improved.
3. The discussion part of the article should include a comparison of the original chip data and discuss the results of your own and the results of other people's chip data.
4. Authors should also investigate the link between the prognosis of collected clinical samples and the results of online analysis.

---

## Round 0.4 · Minor Revisions

Dear Dr. Xiao,

Thank you for providing the revised manuscript. The revised manuscript has been assessed by a reviewer and myself as an Academic Editor. The revised version has certainly been improved. The language of the article, as well as the flow of the writing, has significantly improved. Enclosed are the specific concerns showed by the reviewer regarding the present study. While preparing your revised manuscript, I ask that you submit a detailed response to the reviewer’s comment along with the revised version of the manuscript. These suggestions are essential to incorporate before publication. Finally, I encourage authors to submit a revised version of the manuscript to be considered for final publication in PeerJ.

Best wishes,
Shalu Jhanwar

·

Basic reporting

good

Experimental design

good

Validity of the findings

good

Additional comments

The author's research is very good, the workload is very large, the experimental design is also very good, but there are still some problems:
1. In the method of the abstract, the author has written in the content of the result, which needs to be modified.
2. Why did the author choose CDK1 and CEP55 for follow-up research.
3. In Figure 8, WB results should be analyzed statistically, rather than simply presented.
4. The author collected fewer clinical samples, it would be best if they could be increased and supplemented.
5. In the discussion part, the author wrote more of his own research results. The discussion part should be the analysis and summary of own results and other people's research, rather than the presentation of the summary results.
6. The language of the article needs further improvement.

---

## Round 0.5 · accepted · Accept

Dear Dr. Xiao,

Thank you for providing the revised manuscript and detailed response to reviewer comments. I found the authors have added requested information as well as improved the language of the manuscript. Please consider this manuscript accepted.

Best regards,
Shalu Jhanwar